# Clinicopathologic Characteristics Associated with Prognosis in Ocular Extranodal Marginal Zone B Cell Lymphoma

**DOI:** 10.3390/medicina58060818

**Published:** 2022-06-17

**Authors:** Soyeon Choi, Minjung Seo, Seol Hoon Park, Jae-Cheol Jo, Seoung Wan Chae, Ju-Hyang Lee, Hee Jeong Cha

**Affiliations:** 1Department of Pathology, Ulsan University Hospital, University of Ulsan College of Medicine, Ulsan 05505, Korea; 0734590@uuh.ulsan.kr; 2Department of Nuclear Medicine, Ulsan University Hospital, University of Ulsan College of Medicine, Ulsan 05505, Korea; 0733285@uuh.ulsan.kr (M.S.); seolhoonpark@uuh.ulsan.kr (S.H.P.); 3Department of Hematology and Oncology, Ulsan University Hospital, University of Ulsan College of Medicine, Ulsan 05505, Korea; jcjo@uuh.ulsan.kr; 4Department of Pathology, Kangbuk Samsung Hospital, Sungkyunkwan University School of Medicine, Seoul 03181, Korea; swan.chae@samsung.com; 5Department of Ophthalmology, Ajou University Medical Center, Suwon 16499, Korea

**Keywords:** MALT lymphoma, ocular adnexa, prognostic factor, histology

## Abstract

*Background and Objectives*: Extranodal marginal zone lymphoma of the mucosa-associated lymphoid tissue (MALT) type is the most common subtype of the ocular adnexal lymphoma. Despite its excellent prognosis, some patients experience partial remission or progressive disease. We aimed to evaluate clinicopathologic differences in the treatment responder group by comparing complete remission (CR) and non-complete remission (non-CR). *Materials and Methods*: This study retrospectively reviewed 48 patients who were diagnosed with ocular adnexal MALT lymphoma at Ulsan University Hospital between March 2002 and August 2018. Patients who were followed up for less than 6 months were excluded. Histologic and clinical features were analyzed. The patients were divided into two groups: CR and non-CR. *Results*: Among the 48 patients, 33 achieved CR and 15 achieved non-CR during the median follow-up period of 40.00 months (range, 7–109 months). In univariable analysis, more patients tend to undergo treatment in the CR group, and post-radiotherapy (post-RT) SUVmax, PET and serum lactate dehydrogenase (LDH) levels were higher in the non-CR group (*p* = 0.043, *p* = 0.016, and *p* = 0.042, respectively). In a multivariable analysis, only application of treatment, including radiotherapy or chemotherapy with immunotherapy, was related to CR (odd ratio 7.301, 95% confidence interval 1.273–41.862, *p* = 0.026). In subgroup analysis according to the site of involvement, none of the variables were significant except for the post-RT SUVmax of PET and level of serum LDH in the non-conjunctiva group (*p* = 0.026, and *p* = 0.037, respectively). Seven (14.6%) patients had a recurrence, and those with a recurring site other than the primary site had a higher Ki-67 labeling index, although it was not statistically significant (9.56% vs. 18.00%, *p* = 0.095). *Conclusions*: Although belonging to the early stages, the non-CR rate was high in patients with high serum LDH levels, and recurred patients had higher Ki-67. Thus, considering active treatment is recommended in this group of patients.

## 1. Introduction

Lymphoma is the most common malignancy developing in the ocular adnexa and the third most common malignancy of the conjunctiva [1]. Extranodal marginal zone lymphoma of the mucosa-associated lymphoid tissue (MALT lymphoma) is the most common subtype accounting for around 60% of ocular adnexal lymphomas, followed by follicular lymphoma, diffuse large B-cell lymphoma, and mantle cell lymphoma [2]. The incidences of MALT lymphoma in Asian countries account for 79.5% to 91% compared to Western countries, where it accounts for 35% to 70% [3,4]. The prognosis of lymphoma in the ocular adnexa depends primarily on the histologic subtype [3,5,6]. MALT lymphoma and follicular lymphoma have a better prognosis than that of diffuse large B-cell lymphoma and mantle cell lymphoma [3,6,7]. Despite the excellent prognosis of MALT lymphoma, some patients experience partial remission (PR) or progressive disease (PD).

The overall remission rate in patients with MALT lymphoma treated with doxycycline is 48% [8]. Radiation therapy (RT) is generally used in the early stages with a high local control rate of 85% to 100% [9,10,11]. Chemotherapy, immunotherapy, or clinical trial enrollment is performed in the higher stages. However, the treatment guideline is not established for MALT lymphoma in sites other than the stomach.

To our knowledge, there are few studies on the clinicopathologic prognostic factors associated with ocular adnexal MALT lymphoma. Moreover, since the eighth edition of the American Joint Committee on Cancer tumor-node-metastasis (AJCC/TNM) cancer staging system uses a simplified T staging system compared to that used in the seventh edition, studies proposing prognostic factors are sparse. In this study, we aimed to evaluate the clinicopathologic factors associated with clinical outcomes in these patients.

## 2. Materials and Methods

Forty-eight patients who were diagnosed with ocular adnexal MALT lymphoma from 1 March 2002 to 31 August 2018 at Ulsan University Hospital were retrospectively analyzed. MALT lymphoma was diagnosed using histologic examination. Staging was performed by general physical examination, ophthalmologic examination, serologic test, chest, abdomen and pelvic, and orbital computed tomography (CT), orbital magnetic resonance image (MRI), [18F] fluorodeoxyglucose ([18F]FDG) positron emission tomography (PET), and bone marrow biopsy. All patients were staged according to the 8th edition of the AJCC/TNM cancer staging system (Appendix A). Patients who were followed up for less than 6 months were excluded. This study was approved by the Institutional review board (IRB) of Ulsan University Hospital, and informed consent was waived because only medical records were used (IRB No. 2019-06-009).

### 2.1. Clinical Features

Patients’ clinical demographic information including age, sex, underlying diseases, presenting symptoms, symptom duration, location, localization, TNM stage, treatment modality, radiation dose, treatment response, maximum standardized uptake value (SUVmax) on PET, SUVmax after RT, serologic results including lactate dehydrogenase (LDH), ß2-microglobulin (ß2MG), C-reactive protein (CRP), IgG, and IgG4 levels, IgH gene rearrangement results, follow-up period, and failure-free survival were analyzed. The duration of the symptom is defined as the interval between the first symptoms and the diagnosis. Follow-up period is defined as the time from primary diagnosis until the last follow-up. Failure-free survival is defined as the time from primary diagnosis until the relapse or last follow-up.

### 2.2. Histological Features

All the hematoxylin and eosin slides and immunohistochemistry (IHC) results were reviewed by two pathologists (H.J.C. and S.C.). Dense and diffuse infiltration of neoplastic lymphoid cells were usually present with lymphoepithelial lesions. The patterns that we classified are demonstrated in Figure 1.

#### 2.2.1. Confluent or Nonconfluent

Neoplastic cells forming confluent sheets were regarded as a confluent pattern, and if they were infiltrating in cords or small groups in the soft tissue or in a mixed pattern, they were classified as nonconfluent pattern [12]. 

#### 2.2.2. Plasma Cells

Plasma cells scattered in the neoplastic cells were classified as patchy, and sheets composed of many plasma cells were classified as sheets.

#### 2.2.3. Sclerosis

Intermittent sclerosis was classified as mild, and frequent sclerosis with thick collagen bundles was classified as moderate to severe.

#### 2.2.4. Lymphoepithelial Lesion

More than three destroyed epithelia with lymphocyte exocytosis events were regarded positive.

#### 2.2.5. Large Cell Change

An enlarged nucleus with an irregular nuclear membrane resembling centroblasts or immunoblasts was regarded positive. 

#### 2.2.6. Lymphovascular Invasion

Neoplastic cells invading lymphovascular spaces were regarded positive.

### 2.3. Immunohistochemistry

IHC was performed on 4-µm-thick formalin-fixed paraffin-embedded tissue sections. Polymer method was used with an autostainer BOND-MAX (LEICA, Buffalo Grove, IL, USA). Generally, the expression of CD20, CD3, Bcl-2, CD10, CyclinD1, and Ki-67 was analyzed (Appendix A).

CD20 staining appeared diffuse and strongly in the membranes of neoplastic cells, CD3 staining was visible in reactive T cells, and Bcl-2 staining pattern was similar to that of the CD20. CD10 and CyclinD1 were used to differentiate follicular and mantle cell lymphoma.

#### 2.3.1. Ki-67 Labeling Index

Ki-67 labeling index was assessed by counting the number of tumor cells with nuclear positivity for Ki-67 per 5 × 100 tumor cells using the 40× objective according to the AJCC cancer staging manual. Reactive cells were excluded.

#### 2.3.2. IgG4/IgG Ratio

IgG4+/IgG+ plasma cell ratio of >40% or IgG4 positive plasma cells >50/HPF were regarded positive with IHC [13].

### 2.4. IgH Gene Rearrangement Study

IgH gene rearrangement study was performed using formalin-fixed paraffin-embedded tissues. A 3500 Genetic Analyzer (Applied Biosystem, Waltham, MA, USA) with an IdentiClone^®^ IGH Gene Clonality Assay (Invivoscribe, San Diego, CA, USA) was used. The primer sets were obtained from BIOMED-2 concerted action.

### 2.5. Treatment and Follow-Up

Patients were managed according to disease staging. Tumors confined to the conjunctiva were on observation, including antibiotic therapy and excisional biopsy. All of the patients underwent excisional biopsy, including debulking if needed. RT was performed for the orbital lesion and selectively on the conjunctival lesion, considering the size. Chemotherapy with immunotherapy for advanced stage patients was performed with Rituximab, cyclophosphamide, vincristine, and prednisone (R-CVP) regimen. Rituximab, an anti-CD20 monoclonal antibody, was administered intravenously. Patients were followed-up with ophthalmologic examination every 3 months and the abdomen and pelvic, chest, and orbit CT examination every 6 months. The response evaluation was based on imaging studies and additional slit lamp examination for lesions in the conjunctiva. The patients were classified into four groups depending on the response: CR, disappearance of all evidence of disease; PR, regression of measurable disease and no new sites; stable disease (SD), failure to attain CR/PR or PD; relapsed disease or PD, any new lesion or increase by ≥50% of previously involved site from nadir [14].

### 2.6. Statistical Analysis

Continuous variables were analyzed using the Independent *t*-test or Mann–Whitney U test, and categorical variables were analyzed using the Chi-square test or Fisher’s exact test. Multivariable analysis was performed using logistic regression analysis. All statistical analyses were performed using IBM SPSS Statistics, version 24.0 (IBM Corp., Armonk, NY, USA). All the tests were two-sided, and a *p*-value <0.05 was considered significant.

## 3. Results

### 3.1. Clinicopathologic Characteristics of Patients

The clinical characteristics of patients with ocular adnexal MALT lymphoma are summarized in Table 1. The mean age was 52.08 years old (range 15–75 years), and 27 (56.3%) patients were male. Two (4.2%) had a history of ocular trauma, six (12.8%) had *H.pylori* associated gastritis without MALT lymphoma, and two (4.2%) were positive for hepatitis B. Lid swelling was the most common presenting symptom at the time of diagnosis followed by salmon patch and mass, accounting for 11 (22.9%), 10 (20.8%), and 10 (20.8%) cases, respectively. The mean duration of symptoms was 11.40 months (*n* = 38, range 0–96 months). Twelve (25.0%) cases were bilateral, and conjunctiva (22, 45.8%) was the most common site. Most were in the T1 (22, 45.8%) or T2 (23, 47.9%) stage by the AJCC eighth edition classification. Patients were followed up for a median time of 40.00 months (range 7–109 months). Fifteen patients (31.3%) were on observation after mass excision, 31 (64.6%) received RT, and 2 (4.2%) received chemotherapy with immunotherapy. Tumor response after the initial treatments showed CR (68.8%, *n* = 33), PR (12.5%, *n* = 6), SD (4.2%, *n* = 2), and PD (14.6%, *n* = 7). The mean radiation dose of the RT was 26.64Gy (range 24–36 Gy).

Thirty-three patients were included in the CR group and 15 in the non-CR group. The serum LDH levels were significantly different between the two groups, showing that the CR group was slightly above the cut-off value of 225 U/L and the non-CR group much higher than it (237.33 U/L vs. 290.91 U/L, *p* = 0.042). The mean post-RT SUVmax was higher in the non-CR group, and the patients in the CR group received more active treatment (*p* = 0.016 and *p* = 0.043). 

Microscopically, 10 (20.8%) cases showed a confluent pattern, 31 (64.6%) showed plasma cell infiltration, 32 (66.7%) had sclerosis, and 31 (64.6%) had a lymphoepithelial lesion. Large cell change was observed in four (8.3%) cases, and lymphovascular invasion was observed in most cases (46, 95.8%). The mean value of the Ki-67 index was 10.44% (range 1.0–40.0%). Four cases (8.7%) were positive for IgG4/IgG ratio, which were consistent with the IgG4-related disease criteria described earlier. Fifteen (15/36, 41.7%) cases were positive for monoclonality based on IgH gene rearrangement analysis (Table 2).

According to the Multivariable Analysis, among the LDH Level over 225 U/L, Ki-67 ≥ 5%, treatment including RT and chemotherapy with immunotherapy, and SUVmax on PET, only observation was significantly associated with non-CR (odd ratio 7.301, 95% confidence interval 1.273–41.862, *p* = 0.026) (Table 3). Univariable analyses of the variables included are shown in Appendix A. 

### 3.2. Clinicopathologic Characteristics and Risk Factors for Non-CR in Conjunctiva and Non-Conjunctiva Groups

CR was achieved in 15 (68.2%) patients with MALT lymphoma in the conjunctiva and in 18 (69.2%) with MALT lymphoma in the non-conjunctiva. Post-RT SUVmax and serum LDH level differed between groups in the non-conjunctiva group (*p* = 0.026 and *p* = 0.037, respectively). Furthermore, considering the four recurred patients in the non-conjunctiva group, the Ki-67 labeling index in those who had recurrence was 5, 5, 20, 30%, respectively. In addition, serum LDH levels were higher in the non-CR group than in the CR group in the conjunctiva subgroup, although this difference was not statistically significant (257.13 U/L vs. 291.33 U/L, *p* = 0.483) (Table 4).

### 3.3. Clinicopathologic Characteristics in Recurred Patients

Seven patients (14.6%) experienced recurrence (Table 5). In three cases, lymphoma was primarily in the conjunctiva, and all patients were on observation. All recurrences were in the ipsilateral or contralateral conjunctiva. Two patients received RT, and one received surgery. All achieved CR. Of the four cases of recurrence from the non-conjunctiva group, one case was present diffusely in the left orbit, and after RT, it recurred in both conjunctiva after 35 months. Follow-up was lost since the patient moved to another hospital. The second case of recurrence was diffusely in the left orbit, including the conjunctiva. The patient received RT resulting in CR and had a recurrence in 9 months in the right conjunctiva. RT was performed again, and the patient achieved CR. Two cases were primarily in the lacrimal gland. One patient was previously on immunosuppression therapy for idiopathic thrombocytopenic purpura. She was on observation with doxycycline and had a recurrence in the left eyelid and lung after 35 and 67 months, respectively. Eyelid relapse was treated with doxycycline, and surgery was performed for relapse in the lung resulting in PR. The other patient initially received RT resulting in CR. After 48 months, right cheek recurrence was treated using RT, and CR was achieved. After 26 months, the recurrence was observed in the lung, and a left lower lobe lobectomy was performed. The histology showed transformation into diffuse large B cell lymphoma (DLBCL), the non-germinal center-cell-like type. In five patients who had a recurrence in sites other than the primary site, the Ki-67 labeling indices were higher in the primary slide, although it was not statistically significant (9.56% vs. 18.00%, *p* = 0.095). 

## 4. Discussion

While most ocular adnexal MALT lymphomas are reported to occur in the fifth to seventh decade of life, in the Korean population, they are reported to occur at a significantly younger age at the time of diagnosis, with variable sex predilection [9]. The mean age at the time of diagnosis in our study was 52 years, and a slight predominance of the male sex was observed. The most common symptoms were eyelid swelling, salmon patch, and mass [1,6,15,16]. 

It is a diagnostic challenge to differentiate between reactive lymphoid hyperplasia (RLH) and MALT lymphoma. RLH is composed of polyclonal lymphoid cells, and Dutcher’s bodies and immunoglobulin (Ig) light chain restriction is rarely observed [17]. Most cases in our study show lymphovascular invasion (95.8%), which is an important histologic finding that distinguishes between RLH and MALT lymphoma. Other studies suggest that MALT lymphoma tends to occur in older age, has a predilection for the male sex, and tends to be localized, predominantly in the conjunctiva, compared with RLH [16,18,19]. A certain proportion of malignant transformation from RLH to malignant lymphoma has been reported, which is lower in the conjunctiva. Less aggressive treatment is recommended, such as surgery, steroid, and observation [20]. In this study, we have analyzed the pathologic characteristics of MALT lymphoma in ocular adnexa, which might assist in the diagnosis.

In a study by Lagoo et al., 62 cases of ocular adnexal MALT lymphoma were classified as confluent, infiltrative, and mixed confluent and infiltrative in 74.5, 13.7, and 11.8% of the cases, respectively [12]. In a study by Ferry et al., including 116 cases of ocular adnexal MALT lymphoma, they observed abundant or aggregated plasma cells in 63 cases (35%), Dutcher bodies in 49 cases (27%), sclerosis in 36 cases (20%), and lymphoepithelial lesion in 4 cases (13%) (in 31 cases involving lacrimal gland) [5]. The percentages of cases with plasma cells, sclerosis, and lymphoepithelial lesion were greater in our study because cases with sites other than the lacrimal gland for lymphoepithelial lesions were also included. More lymphoepithelial lesions were observed in the conjunctiva than in sites other than the conjunctiva (95.5% vs. 38.5%, *p* = 0.000). 

Ki-67 labeling index has shown prognostic potential in our study. Only a few other studies have validated the Ki-67 labeling index as a prognostic factor in MALT lymphoma. A study by Petit et al. that included patients with marginal zone B-cell lymphomas and lymphoplasmacytic lymphomas showed that the absence of expression of both Ki-67 and interferon regulatory factor 4 (IRF4 or multiple myeloma oncogene-1-protein, MUM1), were associated with better prognosis [21]. However, in a study by Albano et al., the Ki-67 index was neither associated with progression-free survival nor with overall survival in univariate and multivariate analyses [22]. The limitation concerning the lack of statistical significance may be due to the low Ki-67 index representing its indolent nature or because of the small sample size. However, our study implies that high values should be paid attention to as a warning sign. The remaining problems are how to accurately evaluate the Ki-67 labeling index, excluding reactive cells and its cut-off value. In addition, there is a possibility that we might not be able to evaluate its potential in the slide of the primary site. The Ki-67 indices in patient seven were 10% in the lacrimal gland and 15% in the cheek, but it was increased to 70% with transformation into DLBCL (Figure 2). Another notable finding is that the results of the gene rearrangement analyses showed polyclonality in all three sites. This patient is currently on R-CHOP (Rituximab, Cyclophosphamide, Adriamycin, Vincristine, and Prednisone) chemotherapy. In order to set a cut-off value for Ki-67 index, more studies are necessary, and it must be validated in larger studies.

Although most cases of MALT lymphoma express IgM and develop from a T-helper type 1 (Th1) inflammatory environment, MALT lymphomas expressing other class-switched immunoglobulins, including IgG4, develop from a T-helper type 2 (Th2) inflammatory environment [23]. Several studies have been performed on IgG4-expressing MALT lymphoma among cutaneous lymphomas because of the high proportion of positive cases. In a study by Brenner et al., IgG4 expressing MALT lymphoma showed an excellent prognosis [24]. Sumii et al. showed that the standard treatment for marginal zone B-cell lymphoma is appropriate for IgG4-producing MALT lymphomas in a cases series of seven patients, including five cases of ocular adnexal MALT lymphomas [25]. However, Lee et al. identified that 10% (10/50) of the recruited non-conjunctival ocular adnexal MALT lymphoma patient was positive for IgG4 and showed characteristic histological features such as extensive plasma cell infiltration and dense fibrosis. They found that most of the lymphomas were located in the lacrimal gland (four of five cases) and had a lower response to initial treatment [26]. According to our study, four (8.7%) patients were positive for IgG4, three (75%) had patchy plasma cells infiltration, two (50%) had sclerosis (1 mild and 1 moderate to severe), and two occurred in the lacrimal gland, one in the conjunctiva, and one diffusely. All of these patients received RT and achieved CR. Similarly, in a study by Kubota et al., 10 (9%) were positive for IgG4, 10 (100%) were associated with sclerosis, and 8 (80%) were associated with reactive follicles [27]. However, in a study by Li et al., 30.58% of IgG4/IgG >0.4% cases in MALT lymphoma were reported, and they suggested that IgG and IgG4 might play an important role in the pathogenesis of MALT lymphoma [28]. Some researchers have suggested that MALT lymphoma can develop in the background of IgG4-related disease, and Ohno et al. suggested that IgG4-associated MALT lymphoma may have unique pathogenesis [29,30,31]. Further studies are needed to elucidate the detailed pathogenesis and prognosis of this disease.

Monoclonality was present in only 15 (41.7%) cases. It was consistent with other studies showing that the positive rate was present in 25–70% of cases using polymerase chain reaction with a considerable false-negative rate [18]. Reasons for false-negative rates include limited sensitivity related to normal polyclonal background, clones expressing incomplete rearrangements, or exceedingly rarely, chromosomal translocations involving the Ig locus, or fixation or sampling errors [32]. In our opinion, the negative result of the IgH gene rearrangement study should not preclude the diagnosis of MALT lymphoma, and it should be considered carefully with histology and IHC studies. In addition, it should be noted that the result of the IgH gene rearrangement studies showed polyclonality in all three MALT lymphoma sites, in case6 and case7, including lacrimal glands in both cases and all the metastasized sites. There is a possibility that an association between the result of polyclonality and treatment response exists, which should be studied in the future. 

According to our study, serum LDH levels were higher in the non-CR group in the univariable analysis. According to the Warburg effect, an altered metabolic pathway results in the upregulation of LDH, and it has been studied in many cancer types to show whether it can serve as a diagnostic, prognostic, or predictive marker [33]. LDH is a prognostic factor in International Prognostic Index and is supported by many studies [34,35]. Radiotherapy should be considered in patients with elevated serum LDH levels. In addition, SUVmax (before and after RT) may be another potential prognostic factor [18F]FDG avidity has been reported to be related to the Ki-67 index, plasmacytic differentiation, tumor size, site of the disease, and morphological pattern of presentation in other studies [22,36,37]. The SUVmax and post-RT SUVmax were higher in the non-CR group, and although the sample size was changed, post-RT SUVmax was increased compared to the SUVmax in the non-CR group. Whether [18F]FDG PET is useful in evaluating post-RT effects remains unanswered in our study. According to multivariable analysis, treatments including chemotherapy and RT were important to achieve CR. The effectiveness of RT in localized disease has been proven in numerous studies with excellent outcomes [38,39]. The proposed cut-off doses by others are around 30 Gy [10,11,40,41,42], but cataracts and dry eye syndrome were the main complications in our study. In our study, only two patients were on chemotherapy, and a few were in advanced stages; therefore, the conclusion cannot be made. Two patients in stage T3 had lymphoma diffusely in orbit and received RT, which resulted in CR. One patient in stage T4 had lymphoma in the left orbit, both sides of the neck, left axilla, and right iliac lymph nodes, and was confirmed with orbital mass excision. He had eight cycles of chemotherapy with immunotherapy, which resulted in partial response. However, chemotherapy with Rituximab is recommended for advanced stages [39].

In an article by Jung et al. concerning conjunctival lymphoproliferative lesions in South Korea, bilaterality was present in 41.7% (50/120), which was much higher than that reported in other studies. They also discovered that bilaterality was more common in follicular appearance in slit-lamp examination than in salmon-patch appearance. Considering that follicular appearance was a predominant clinical presentation in chronic inflammation according to their studies, the authors suggested that follicular appearance reflected an extension from chronic inflammation, and they were more likely to be bilateral [43]. Chronic antigenic stimulation has been postulated as an etiology in MALT lymphoma, and *Chlamydia psittaci* (Cp) has been known to be the causative pathogen of the ocular adnexa. The reported Cp positive rates were variable geographically, from 0% to 87% [44]. Based on these studies, it can be suggested that chronic inflammation precedes MALT lymphoma. In addition, in accordance with this previous publication, bilaterality was prevalent in conjunctival MALT lymphoma compared to non-conjunctiva MALT lymphoma in our study (40.9% vs. 11.5%., *p* = 0.042), but it was not associated with prognosis.

Several limitations are present. First, there was a bias associated with its retrospective design. Second, a small number of cases were analyzed because of the rare prevalence of ocular adnexa MALT lymphoma. Third, sampling error or reproducibility can be a concern. However, a careful examination of the available data was performed, useful images were acquired, and for the Ki-67 index, two pathologists separately counted the positive cells and reached an agreement. Care was taken to exclude reactive cells, including the germinal center.

The strength of our study is that it analyzed the relevant data on clinicopathologic characteristics of patients with ocular adnexal MALT lymphoma, which is a rare disease, in a single tertiary institution with a sufficient follow-up period.

## 5. Conclusions

In conclusion, active treatment is recommended even in the early stages of ocular adnexal MALT lymphoma, especially when the Ki-67 index is high and serum LDH level is higher than the normal range. Our study supports radiotherapy, chemotherapy or immunotherapy is essential in achieving CR, and since LDH level was associated with non-CR, more caution should be paid with active treatment. Additionally, for accurate diagnosis, the polyclonal result seen in the IgH gene rearrangement study should not be solely considered to exclude the diagnosis of MALT lymphoma; the result should be interpreted in combination with other findings. This will help accurately diagnose ocular adnexal MALT lymphoma and investigate related predictive and prognostic factors.

## Figures and Tables

**Figure 1 medicina-58-00818-f001:**
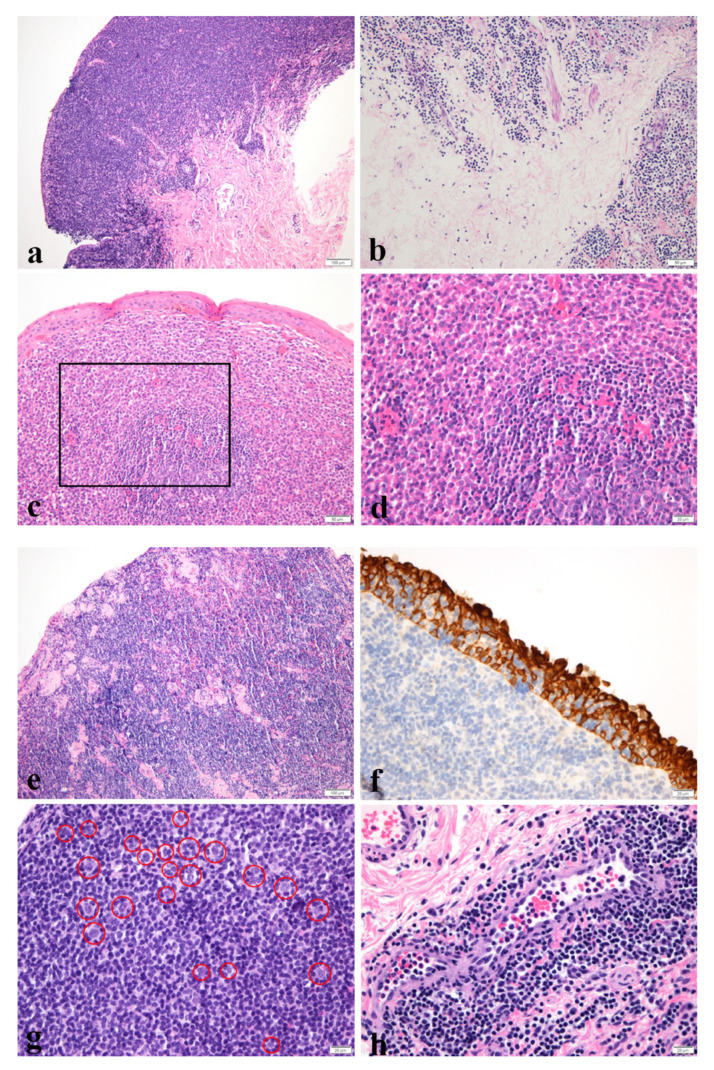
Images show pathological characteristics. (**a**) Neoplastic cells are infiltrating in a confluent pattern. (H&E, ×100), (**b**) Neoplastic cells are infiltrating in a nonconfluent pattern. (H&E, ×200), (**c,d**) Tumor cells are differentiated into sheets of plasma cells. d represents the magnification view of the inlet box of **c**. (H&E, **c**; ×200, **d**; 400), (**e**) Multifocal sclerosis is intermixed with neoplastic lymphoid cells. (H&E, ×100), (**f**) Immunohistochemistry staining for CK reveals lymphoepithelial lesion. (CK, ×400) (**g**) Medium-sized lymphoid cells are intermixed with cells transformed into large cells. The large cells are indicated with red circles. (H&E, ×400), (**h**) High power field view shows lymphovascular invasion. (H&E, ×400). H&E, hematoxylin, and eosin; CK, cytokeratin.

**Figure 2 medicina-58-00818-f002:**
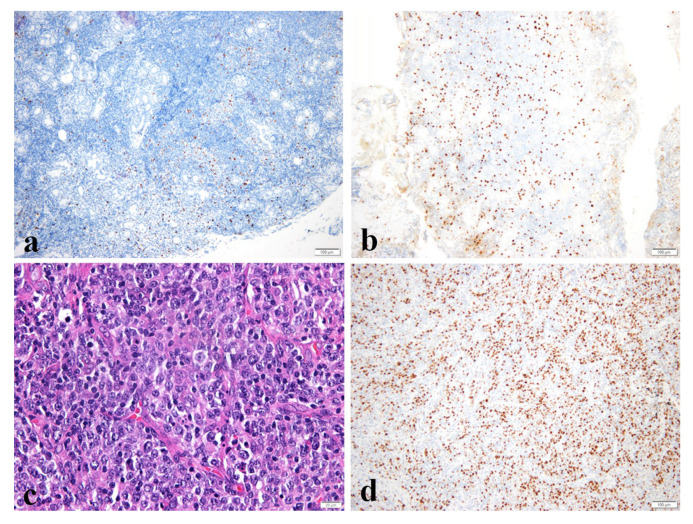
Images of Ki-67 index in case6, and diffuse large B cell lymphoma transformed case7. (**a**,**b**) Immunohistochemistry staining for Ki-67 shows nuclear positivity in 5% of tumor cells in the initial slide but is increased to 30% in the recurred slide in eyelid. (Ki-67, a; ×100, b; ×100), (**c**,**d**) Transformed cells are scattered between small lymphoid cells and Ki-67 staining shows increment. (H&E, c; ×400, Ki-67, d; ×100).

**Table 1 medicina-58-00818-t001:** Comparison of clinical characteristics in ocular adnexal MALT lymphoma between complete remission group and non-complete remission group.

Variable		Response, *n* (%)		*p*-Value
	Total (*n* = 48)	CR (*n* = 33)	Non-CR (*n* = 15)	
Age	52.08 (14.95)	52.36 (14.60)	51.47 (16.19)	0.850
Sex, *n* (%)				0.367
Male	27 (56.3%)	20 (60.6%)	7 (46.7%)	
Female	21 (43.8%)	13 (39.4%)	8 (53.3%)	
Laterality, *n* (%)				0.857
Unilateral	36 (75.0%)	25 (75.8%)	11 (73.3%)	
Bilateral	12 (25.0%)	8 (24.2%)	4 (26.7%)	
T stage				1.000ᵇ
T1	22 (45.8%)	15 (45.5%)	7 (46.7%)	
T2	23 (47.9%)	15 (45.5%)	8 (53.3%)	
T3	2 (4.2%)	2 (6.1%)	0 (0%)	
T4	1 (2.1%)	1 (3.0%)	0 (0%)	
Management				0.043 ᵇ*
Observation	15 (31.3%)	7 (21.2%)	8 (53.3%)	
Treatment	33 (66.8%)	26 (78.8%)	7 (46.7%)	
Radiotherapy	31 (64.6%)	24 (72.7%)	7 (46.7%)	
Chemotherapy	2 (4.2%)	2 (6.1%)	0 (0%)	
RT dose (Gy) (*n* = 23, 7)	26.64 (3.38)	26.61 (3.54)	26.74 (3.08)	0.635 ᵃ
SUV_max_ (*n* = 32, 12)	2.55 (1.42)	2.52 (1.30)	2.64 (1.78)	0.822 ᵃ
SUV_max_ (post-RT) (*n* = 14, 4)	2.27 (0.99)	1.98 (0.77)	3.27 (1.13)	0.016 *
LDH (U/L) (*n* = 33, 13)	252.48 (82.65)	237.33 (75.51)	290.91 (90.44)	0.042 ᵃ*
ß2MG >2.19 (mg/L) (*n* = 31, 13)	6 (13.6%)	5 (16.1%)	1 (7.7%)	0.457
CRP >0.50 (mg/dL) (*n* = 33, 13)	2 (4.3%)	1 (3.0%)	1 (7.7%)	0.490 ᵇ
Serum IgG4 >135 (mg/dL)(*n* = 22, 6)	7 (25.0%)	4 (18.2%)	3 (50.0%)	0.144 ᵇ
Failure-free survival (months)	38.00 (21.67)	40.58 (21.83)	32.33 (20.93)	0.222
Follow-up period (month)	43.33 (24.35)	40.58 (21.83)	49.40 (29.06)	0.249

* *p* < 0.05, ᵃ Mann–Whitney U test, ᵇ Fisher’s exact test, Values are presented with *n* (%) or mean (standard deviation). The differed numbers of samples for each CR and non-CR are provided in the parenthesis after the variable name. CR, complete remission; Gy, gray; SUVmax, maximum standardized uptake value; RT, radiotherapy; LDH, lactate dehydrogenase; ß2MG, ß2-microglobulin; CRP, C-reactive protein.

**Table 2 medicina-58-00818-t002:** Comparison of pathologic characteristics in complete remission group versus non-complete remission group in ocular adnexal MALT lymphoma.

Variable		Response, *n* (%)		*p*-Value
	Total (*n* = 48)	CR (*n* = 33)	Non-CR (*n* = 15)	
Histologic pattern				0.151
Confluent	10 (20.8%)	5 (15.2%)	5 (33.3%)	
Nonconfluent	38 (79.2%)	28 (84.8%)	10 (66.7%)	
Plasma cells				0.531
Absent	17 (35.4%)	13 (39.4%)	4 (26.7%)	
Patchy	20 (41.7%)	12 (36.4%)	8 (53.3%)	
Sheet	11 (22.9%)	8 (24.2%)	3 (20.0%)	
Sclerosis				0.306
Absent	16 (33.3%)	9 (27.3%)	7 (46.7%)	
Mild	12 (25.0%)	10 (30.3%)	2 (13.3%)	
Moderate to severe	20 (41.7%)	14 (42.4%)	6 (40.0%)	
Lymphoepithelial lesion	31 (64.6%)	20 (60.6%)	11 (73.3%)	0.393
Large cell change	4 (8.3%)	2 (6.1%)	2 (13.3%)	0.579 ᵇ
Lymphovascular invasion	46 (95.8%)	31 (93.9%)	15 (100.0%)	1.000 ᵇ
Ki-67	10.44 (10.69)	9.40 (9.53)	11.53 (13.21)	0.588 ᵃ
IgG4/IgG ≥ 0.4 (*n* = 33, 13)	4 (8.7%)	4 (12.1%)	0 (0.0%)	0.313 ᵇ
IgH rearrangement (*n* = 26, 10)				0.900
Monoclonal	15 (41.7%)	11 (43.3%)	4 (40.0%)	
Polyclonal	21 (58/3%)	15 (57.7%)	6 (60.0%)	

ᵃ Mann–Whitney U test, ᵇ Fisher’s exact test, Values are presented with *n* (%) or mean (standard deviation). The differed numbers of samples for each CR and non-CR are provided in the parenthesis after the variable name. CR, complete remission; MALT, mucosa-associated lymphoid tissue.

**Table 3 medicina-58-00818-t003:** Multivariable analysis of clinicopathologic factors regarding the non-complete remission outcome (*n* = 43).

Variable	Odd Ratio	95% CI	*p*-Value
LDH >225 U/L	4.142	0.807–21.269	0.089
Ki-67 ≥ 5%	0.678	0.133–3.447	0.639
Treatment (RT or CT with immunotherapy)	7.301	1.273–41.862	0.026 *
SUV_max_	1.349	0.769–2.365	0.296

CI, confidence interval; LDH, lactate dehydrogenase; RT, radiotherapy; CT, chemotherapy; SUVmax, maximum standardized uptake value; * *p* < 0.05; Nagelkerke R2 = 0.284.

**Table 4 medicina-58-00818-t004:** Comparison of clinicopathologic characteristics between conjunctival and non-conjunctival MALT lymphoma.

	Conjunctival MALT Lymphoma (*n* = 22)		Non-Conjunctival MALT Lymphoma (*n* = 26)			
Variable	CR (*n* = 15)	Non-CR (*n* = 7)	*p*-Value	CR (*n* = 18)	Non-CR (*n* = 8)	*p*-Value
Histologic pattern			0.145 ᵇ			1.000 ᵇ
Confluent	3 (20.0%)	4 (57.1%)		2 (11.1%)	1 (12.5%)	
Nonconfluent	12 (80.0%)	3 (42.9%)		16 (88.9%)	7 (87.5%)	
Plasma cells			0.310 ᵇ			0.704 ᵇ
Absent	5 (33.3%)	0 (0.0%)		8 (44.4%)	4 (50.0%)	
Patchy	5 (33.3%)	4 (57.1%)		7 (38.9%)	4 (50.0%)	
Sheet	5 (33.3%)	3 (42.9%)		3 (16.7%)	0 (0.0%)	
Sclerosis			0.165 ᵇ			0.856 ᵇ
Absent	6 (40.0%)	5 (71.4%)		3 (16.7%)	2 (25.0%)	
Mild	6 (40.0%)	0 (0.0%)		4 (22.2%)	2 (25.0%)	
Moderate to severe	3 (20.0%)	2 (28.6%)		11 (61.1%)	4 (50.0%)	
Lymphoepithelial lesion	14 (93.3%)	7 (100.0%)	1.000 ᵇ	6 (33.3%)	4 (50.0%)	0.664 ᵇ
Large cell change	0 (0%)	2 (28.6%)	0.091 ᵇ	2 (11.1%)	0 (0.0%)	1.000 ᵇ
Lymphovascular invasion	13 (86.7%)	7 (100.0%)	1.000 ᵇ	18 (100.0%)	8 (100.0%)	
Ki-67	9.73 (12.08)	15.00 (15.73)	0.722 ᵃ	10.11 (7.12)	8.50 (10.69)	0.221 ᵃ
IgG4/IgG ≥ 0.4 (*n* = 15, 6 and *n* = 18, 7)	1 (6.3%)	0 (0%)	1.000 ᵇ	3 (16.7%)	0 (0.0%)	0.534 ᵇ
Laterality			0.376 ᵇ			0.529 ᵇ
Unilateral	10 (66.7%)	3 (42.9%)		15 (83.3%)	8 (100.0%)	
Bilateral	5 (33.3%)	4 (57.1%)		3 (16.7%)	0 (0.0%)	
Management			0.165 ᵇ			0.086 ᵇ
Observation	7 (46.7%)	6 (85.7%)		0 (0.0%)	2 (25.0%)	
Treatment	8 (53.3%)	1 (14.3%)		18 (100.0%)	6 (75.0%)	
Radiotherapy	8 (53.3%)	1 (14.3%)		16 (88.9%)	6 (75.0%)	
Chemotherapy	0 (0.0%)	0 (0.0%)		2 (11.1%)	0 (0.0%)	
SUV_max_ (*n* = 14, 6 and *n* = 18, 6)	1.86 (0.37)	1.75 (0.25)	0.533	3.03 (1.52)	3.53 (2.24)	0.617 ᵃ
SUV_max_ (post-RT) (*n* = 3, 0 and *n* = 11, 4)	1.23 (1.07)			2.18 (0.58)	3.27 (1.13)	0.026 *
LDH (U/L) (*n* = 15, 6 and *n* = 18, 7)	257.13 (94.62)	291.33 (110.49)	0.483ᵃ	220.83 (52.20)	290.57 (78.64)	0.037 ᵃ*
ß2MG > 2.19 (mg/L) (*n* = 13, 6 and *n* = 18, 7)	3 (23.1%)	0 (0.0%)	0.200	2 (11.1%)	1 (14.3%)	0.826
Failure-free survival (month)	44.87 (28.63)	30.00 (24.68)	0.252	37.00 (13.85)	34.38 (18.55)	0.691
Follow-up period (month)	44.87 (28.12)	42.71 (28.12)	0.871	37.00 (13.85)	55.25 (30.46)	0.142

* *p* < 0.05, ᵃ Mann–Whitney U test, ᵇ Fisher’s exact test, Values are presented with *n* (%) or mean (standard deviation). The differed numbers of samples for each CR and non-CR are provided in the parenthesis after the variable name. CR, complete remission; CRP, C-reactive protein; LDH, lactate dehydrogenase; MALT, mucosa-associated lymphoid tissue; RT, radiotherapy; SUVmax, maximum standardized uptake value.

**Table 5 medicina-58-00818-t005:** Clinical characteristics in recurred patients.

Patient #	Age/Sex	Primary Site	Treatment	Time to Relapse (Month)	Recur Site	Treatment	Outcome
1	32/M	Right conjunctiva	Observation (Antibiotics)	80	Right conjunctiva	Radiation	CR
2	64/F	Left conjunctiva	Observation (Antibiotics)	27	Right conjunctiva	Observation (Excision)	CR
3	35/F	Right conjunctiva	Observation (Excision)	6	Right conjunctiva	Radiotherapy	CR
4	56/F	Left orbit	Radiotherapy	35	Both conjunctiva	NA	NA
5	74/F	Left orbit	Radiotherapy	9	Right conjunctiva	Radiotherapy	CR
6	47/F	Right lacrimal gland	Observation (Antibiotics)	35 and 67	Left eyelid and lung (multiple)	Obervation (Antibiotics) and Surgery	PR
7	65/F	Right lacrimal gland	Radiotherapy	48 and 26	Right cheek and lung (DLBCL transformation)	Radiotherapy and surgery with chemotherapy and immunotherapy	CR

M, male; F, female; CR, complete remission; NA, not available; PR, partial remission; DLBCL, diffuse large B cell lymphoma.

## Data Availability

The data presented in this study are available on request from the corresponding author.

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
