# Peer review of "Clinicopathologic Characteristics Associated with Prognosis in Ocular Extranodal Marginal Zone B Cell Lymphoma"

_medicina, 2022, doi:10.3390/medicina58060818_

Round 1
Reviewer 1 Report
This is a well-conceived and interesting study on ocular extranodal marginal zone B-cell lymphoma. The main finding is that active treatment with radiation therapy or chemotherapy is associated with a higher rate of CR. Higher SUVmax after radiation therapy and high LDH levels are associated with non-CR in the non-conjonctiva group. Finally, higher Ki-67 was associated with a risk of relapse.
My comments:
- It would be interesting to have more details on the chemotherapy used. What were the chemotherapies administrated?
- What is the mean duration of symptoms? Is it the duration between the first symptoms and the diagnosis? Or the duration between the first symtoms and the end of the symptoms after treatment?
- Lines 177 to 179: these two sentences could be deleted.
- In Table 1, the Ki67 could be added.
- It is not clear if all patients had mass excision or not. The “observation group” seem to have had mass excision but it is not clear if other patients had surgery. It would seem more clear to acknowledge that surgery is not enough and that all patients should receive radiation therapy or chemotherapy to enhance the chances of CR.
- What about immunotherapy with anti-CD20 monoclonal antibody? Did any patient receive this therapy with subcutaneous or intravenous injection? What about intraocular injections? This should be discussed.
- It would be interesting to have the details of the multivariate analyses. A table could be added in the supplementary data. Why Ki-67 index was included in the multivariate analysis?
Author Response
1) It would be interesting to have more details on the chemotherapy used. What were the chemotherapies administrated?
Response 1)
Two patients were treated with chemotherapy and immunotherapy. Both patients were treated with Rituximab, cyclophosphamide, vincristine, and prednisone (R-CVP) regimen. Additionally, the DLBCL transformed case is being treated with Rituximab, cyclophosphamide, adriamycin, vincristine, and prednisone (R-CHOP) chemotherapy. According to the hematology and oncology professor in our institution, the insurance doesn’t cover R-CHOP chemotherapy in our country, so we are treating patients with R-CVP.
2) What is the mean duration of symptoms? Is it the duration between the first symptoms and the diagnosis? Or the duration between the first symtoms and the end of the symptoms after treatment?
Response 2)
The duration of the symptom is defined as the interval between the first symptoms and the diagnosis. Following is added in the Materials and Methods section; The duration of the symptom is defined as the interval between the first symptoms and the diagnosis. Thank you.
3) Lines 177 to 179: these two sentences could be deleted.
Response 3)
Thank you. I have deleted.
4) In Table 1, the Ki67 could be added.
Response 4)
Ki-67 labeling index is in the Table2 where the pathological characteristics are compared. (10.44 (10.69), 9.94 (9.53), 11.53 (13.21) p=0.588ᵃ; mean(SD); total, CR , non-CR, and p value, respectively)
5) It is not clear if all patients had mass excision or not. The “observation group” seem to have had mass excision but it is not clear if other patients had surgery. It would seem more clear to acknowledge that surgery is not enough and that all patients should receive radiation therapy or chemotherapy to enhance the chances of CR.
Response 5)
For the lesions in conjunctiva, we performed excisional biopsy trying to debulk most of the parts, and for smaller lesions we performed excisional surgery including biopsy followed by observation. If the size was large, we recommended radiotherapy. For orbital lesions, we performed either radiotherapy or chemotherapy with immunotherapy after biopsy.
Following is added in the conclusion.; Our study supports radiotherapy, chemotherapy, or immunotherapy is essential in achieving CR.
6) What about immunotherapy with anti-CD20 monoclonal antibody? Did any patient receive this therapy with subcutaneous or intravenous injection? What about intraocular injections? This should be discussed.
Response 6)
All of the immunotherapies were done in intravenous route with anti-CD20 monoclonal antibody in our study. Following was added in the treatment and follow-up section in Materials and Methods.; Chemotherapy with immunotherapy for advanced stage patients was performed with Rituximab, cyclophosphamide, vincristine, and prednisone (R-CVP) regimen. Rituximab, an anti-CD20 monoclonal antibody was administered intravenously.
7) It would be interesting to have the details of the multivariate analyses. A table could be added in the supplementary data. Why Ki-67 index was included in the multivariate analysis?
Response 7)
The multivariable analysis Table3 is replaced according to the second reviewer’s comment. Since the age factor in IPI is not very significant in low grade lymphoma, it is removed. LDH and Ki-67 is change to normal/elevated. Ki-67 cutoff value of 5% is the median in this study and cutoff value of 225U/L for LDH is referred from American board of internal medicine.
Ki-67 labeling index and the SUVmax of PET are important factors in diagnosing high grade lymphoma. So although they were not significant in the univariable analyses, we included them in the multivariable analysis. In addition, although not significant, Ki-67 is higher in the non-CR group in the table 2, and in conjunctival subgroup in table 4 (9.40(9.53), 11.53 (13.21), p=0.588ᵃ and 9.73 (12.08), 15.00 (15.73), p=0.722ᵃ; mean(SD), CR, non-CR, and p-value, respectively).
Supplementary table3 is added for multivariable analysis.
We very thank you for your advice.
Reviewer 2 Report
This is an interesting paper describing the experience from a single institution.
The number of tables in this paper is above average, and you may consider reducing Table 4 and move some of the details to supplementary material.
Table 1: It is unclear what the meaning of the number in parenthesis represent. I assume that is is the number of patients in the CR group and in the non-CR group, but it has to be clarified and clear to the reader.
Table1: For LDH and B2MG,it is unclear what the value represents. Is it mean, median ? If so, please clarify, but what doses the number in the parenthesis represent, Ie 252.48 (82.65) is it deviation ? I would suggest that a cut-off value is selected for LDH and B2MG.
Table 3: LDH and KI-67, I suggest that you use normal/elevated instead of continuous values.
Line 167: “Patients were followed up for a mean time of 43.33 months”. Please use median follow-up time.
Table 5: Information regarding time of relapse (month after primary diagnosis) should be added to the table
LDH: In conclusion (line 383) it is stated that High LDH lvels are of prognostic importance. The term "High LDH " is not very precise, and in line 173 : "that below the normal cut-off value of 280U/L and the non-CR group slightly above (237.33U/L vs. 290.91U/L)"
Line 243-248. The population is somewhat different from other reports. Could you elaborate on this matter, there must be some reason for this ?
Although the number is low the prognostic value of elevated LDH should be implemented and thereby making the statement clearer when you can show that patients with elevated LDH should be treated.
Author Response
1) The number of tables in this paper is above average, and you may consider reducing Table 4 and move some of the details to supplementary material.
Response 1)
IgH rearrangement. RT dose, CRP elevation, Serum IgG4 levels are deleted from Table4.
2) Table 1: It is unclear what the meaning of the number in parenthesis represent. I assume that is is the number of patients in the CR group and in the non-CR group, but it has to be clarified and clear to the reader.
Response 2)
Following is noted for each table below; The differed numbers of samples for each CR and non-CR are given in the parenthesis after the variable name.
3) Table1: For LDH and B2MG,it is unclear what the value represents. Is it mean, median ? If so, please clarify, but what doses the number in the parenthesis represent, Ie 252.48 (82.65) is it deviation ? I would suggest that a cut-off value is selected for LDH and B2MG.
Response 3)
They are in mean (standard deviation) value in every table and the numbers is the parenthesis are specified below the tables.
The cutoff value of LDH is 225U/L from American Board of Internal Medicine, and b2Mg is 2.19 mg/L. The cutoff value was used in multivariable analysis for LDH in Table 3 and b2MG value is changed in Table 1 and 4.
4) Table 3: LDH and KI-67, I suggest that you use normal/elevated instead of continuous values.
Response 4)
It is corrected and Table3 is replaced. The LDH cutoff value is set as 225U/L and the Ki-67 index cutoff value is set according to its median in our study.; (5%)
5) Line 167: “Patients were followed up for a mean time of 43.33 months”. Please use median follow-up time.
Response 5)
Median follow up time is 40months (IQR 24.75-58.25).
6) Table 5: Information regarding time of relapse (month after primary diagnosis) should be added to the table
Response 6)
Additional information was added including the time of relapse in Table5 and also in the text.
7) LDH: In conclusion (line 383) it is stated that High LDH lvels are of prognostic importance. The term "High LDH " is not very precise, and in line 173 : "that below the normal cut-off value of 280U/L and the non-CR group slightly above (237.33U/L vs. 290.91U/L)"
Response 7)
Line 173 is changed as following.; The serum LDH levels were significantly different between the two groups, with showing that the CR group slightly above the cutoff value of 225 U/L and the non-CR group much higher than it (237.33U/L vs. 290.91U/L, p=0.042).
Line 383 is changed as following.; In conclusion, active treatment is recommended even in the early stages of ocular adnexal MALT lymphoma, especially when the Ki-67 index is high and serum LDH level is higher than normal range.
8) Line 243-248. The population is somewhat different from other reports. Could you elaborate on this matter, there must be some reason for this ?
Response 8)
Stefanovic et al. stated OAMLs are mostly seen in the 5th to 7th decade of life (median age, ~65 years), with female predominance (male/female = 1:1.5/2). In contrast, studies in Korean populations reveal a significantly younger age (median, ~46 years) at the time of diagnosis, with male rather than female predominance.(11,16,17)
However in the first reference, Cho et al., (Clinicopathologic Analysis of Ocular Adnexal Lymphomas: Extranodal Marginal Zone B-Cell Lymphoma Constitutes the Vast Majority of Ocular Lymphomas Among Koreans and Affects Younger Patients, American Journal of Hematology 73:87–96 (2003)© 2003 Wiley-Liss, Inc. ) 61 primary and secondary ocular adnexal MALT lymphoma(OAL) patients were included and the M/F ratio is 25:36 in table1.
In the second reference, Woo et al., (The Clinical Characteristics and Treatment Results of
Ocular Adnexal Lymphoma, Korean J Ophthalmol Vol.20, No.1, 2006) of the 15 patients with OAL, the M/F ratio was 10:5 according to table1.
In the third reference, Yoon et al., (Prognosis for patients in a Korean population with ocular adnexal lymphoproliferative lesions, Ophthalmic Plast Reconstr Surg. Mar-Apr 2007;23(2):94-9. doi: 10.1097/IOP.0b013e318030b058.) of the 69 patients, 60 OAL patients were included, and the M/F ratio was 40/29 in total.
I stated variable sex distribution.
Concerning the symptoms, the first reference by Shields et al. analyzed only conjunctival lesions including lymphoma. It states “Conjunctival lymphoma classically presents as a pink, salmon-colored subconjunctival mass in the substantia propria sometimes with feeder vessels (Fig. 6). This smooth, multilobulated mass can resemble follicular or papillary conjunctivitis. … The main symptoms include a mass (30%), irritation(29%), ptosis (8%), epiphora (7%), blurred vision (5%), proptosis (3%), diplopia (3%), or no symptoms (15%).63 In addition to the conjunctival infiltration, lymphoma can be found infiltrating the orbit (15%), eyelid (3%), or uvea (4%).63 Based on the current literature, there are no features that differentiate conjunctival lymphoma into histopathologic subtypes.”
In the second reference by Kirkgarrd et al. the symptoms of conjunctival MALT lymphoma were as following; Tumor/swelling(112, 90%), Exophthalmus (33, 27%), Epiphora (14, 11%), Ptosis (13, 10%), Decreased VA (6, 5%), Dry eye (0), Diplopia (1, 1%), B-symptoms (5, 4%), not stated (56, 31%). Signs were as following; Mass (112, 90%), Exophthalmus (22, 17%), Chemosis (22, 17%), Eyeball displacement (20, 16%), restricted eye movement (13, 10%), Edema (13, 10%), Epiphora (8, 6%), Ptosis (7, 6%), Diplopia (4, 3%), Resistance (0), Not state (56, 31%).
In the third reference by Kwon et al., of the 140 ocular adnexal lymphoma patients, 129 MALT patients were included, and the most common presenting symptom overall was palpable/visible mass (56.4%, n=79).
In the forth reference by Asadi-Amoli et al., analyzed 110 ocular adnexal lymphoproliferative lesions with 19 MALT lymphoma, including 2 conjunctival cases. The chief compaints were as following; eye swelling/mass 47(43.1%), Visual disturbance 2(1.8%), Lacrimal gland swelling 3(2.8%), Pain/redmess 1(0.9%), Undetermined 56(51.4%)
Our analysis is below.
|
Presenting symptom |
Total (n=48) |
Conjunctiva (n=22) |
Non-conjunctiva (n=26) |
|
Proptosis |
3 (6.3%) |
0 |
3 (11.5%) |
|
Proptosis and eyelid swelling |
1 (2.1%) |
0 |
1 (3.8%) |
|
Proptosis and epiphora |
1 (2.1%) |
0 |
1 (3.8%) |
|
Eyelid swelling |
8 (16.7%) |
0 |
8 (30.8%) |
|
Eyelid swelling and pain |
1 (2.1%) |
0 |
1 (3.8%) |
|
Eyelid swelling and visual loss |
1 (2.1%) |
0 |
1 (3.8%) |
|
Palpable mass |
9 (18.8%) |
4 (18.2%) |
5 (19.2%) |
|
Palpable mass and diplopia |
1 (2.1%) |
0 |
1 (3.8%) |
|
Salmon patch |
10 (20.8%) |
7 (31.8%) |
3 (11.5%) |
|
Epiphora |
5 (10.4%) |
3 (13.6) |
2 (7.7%) |
|
Ptosis |
1 (2.1%) |
1 (4.5%) |
0 |
|
Foreign body sense |
4 (8.3%) |
4 (18.2%) |
0 |
|
Follicle |
3 (6.3%) |
3 (13.6%) |
0 |
Our results were consistent with other literatures in that salmon patch 7(31.8%) was the most common symptom in conjunctival MALT lymphoma and eyelid swelling 11(42.2%) was the most common symptom in non-conjunctival MALT lymphoma. Palpable mass 10(20.9%) was also another prevalent symptom overall. There are too many tables so I did not include it.
9) Although the number is low the prognostic value of elevated LDH should be implemented and thereby making the statement clearer when you can show that patients with elevated LDH should be treated.
Response 9)
Following is added in the conclusion : and since LDH level was associated with non-CR, more caution should be paid with active treatment.
We thank you very much for your advice.
Reviewer 3 Report
Minor English review needed
examples
line 46 follicular lymphoma, diffuse large B cell lymphoma and mantle cell lymphoma.
line 139: Patients were managed according to disease staging.
Please review and may be remove line 177-179"This section may be divided by subheadings. It should provide a concise and precise description of the experimental results, their interpretation, as well as the ex perimental conclusions that can be drawn"
line 198 please explain what do you mean by "only appliance of treatment"
line which should be studied in the future
Author Response
1) line 46 follicular lymphoma, diffuse large B cell lymphoma and mantle cell lymphoma.
2) line 139: Patients were managed according to disease staging.
3) Please review and may be remove line 177-179"This section may be divided by subheadings. It should provide a concise and precise description of the experimental results, their interpretation, as well as the ex perimental conclusions that can be drawn"
Response 1-3)
They are corrected. Thank you.
4) line 198 please explain what do you mean by "only appliance of treatment"
Response 4)
It means treatment including RT and chemotherapy with immunotherapy described earlier. It is changed to "application of treatment".
5) line which should be studied in the future
Response 5)
In the discussion, first, we have proposed studies with larger population for determining cut-off value for Ki-67 labeling index for ocular adnexal MALT lymphoma for prognostication. Although in our study the Ki-67 labeling index was not statistically significant when compared the patients’ which occurred in other than primary site (case 2, 4, 5, 6, 7), and the others (9.56% vs. 18.00%, p=0.095). Moreover, in table 2 and 4 in the entire samples and in conjunctiva group, Ki-67 labeling indexes are higher in the non-CR group (9.40(9.53), 11.53 (13.21), p=0.588ᵃ and 9.73 (12.08), 15.00 (15.73), p=0.722ᵃ; mean(SD), CR, non-CR, and p-value, respectively).
Second, IgG4 expressing MALT lymphoma seems to have unique pathologic characteristics and pathogenesis. So, the detailed studies on pathogenesis or prognosis might be help to decide treatment.
Third, in our study, two cases with distant metastases had polyclonal results in IgH rearrangement studies in all the 3 sites each, although the sensitivity was not good. Studies revealing the reasons might help.
Fourth, although many studies are being reported, the usage of [18F]FDG PET for MALT lymphoma can be studied.
Fifth, the association of plasmacytic differentiation with prognosis can be studied since it is not concluded, yet.
We very thank you for all the advice.